https://doi.org/10.1038/s41467-019-12164-y · **OPEN**

# Distinct initiating events underpin the immune and metabolic heterogeneity of *KRAS*-mutant lung adenocarcinoma

Sarah A. Best [1,2], Sheryl Ding[1,2], Ariena Kersbergen[1], Xueyi Dong[2,3], Ji-Ying Song[4], Yi Xie[3,5], Boris Reljic[6], Kaiming Li [1,7], James E. Vince[2,8], Vivek Rathi[9], Gavin M. Wright[10], Matthew E. Ritchie [3,11] & Kate D. Sutherland [1,2]

The *KRAS* oncoprotein, a critical driver in 33% of lung adenocarcinoma (LUAD), has remained an elusive clinical target due to its perceived undruggable nature. The identification of dependencies borne through common co-occurring mutations are sought to more effectively target *KRAS*-mutant lung cancer. Approximately 20% of *KRAS*-mutant LUAD carry loss-of-function mutations in *KEAP1*, a negative regulator of the antioxidant response transcription factor NFE2L2/NRF2. We demonstrate that *Keap1*-deficient *Kras*^G12D lung tumors arise from a bronchiolar cell-of-origin, lacking pro-tumorigenic macrophages observed in tumors originating from alveolar cells. *Keap1* loss activates the pentose phosphate pathway, inhibition of which, using 6-AN, abrogated tumor growth. These studies highlight alternative therapeutic approaches to specifically target this unique subset of *KRAS*-mutant LUAD cancers.

[1] Cancer Biology and Stem Cells Division, The Walter and Eliza Hall Institute of Medical Research, Parkville, VIC 3052, Australia. [2] Department of Medical Biology, The University of Melbourne, Parkville, VIC 3052, Australia. [3] Epigenetics and Development Division, The Walter and Eliza Hall Institute of Medical Research, Parkville, VIC 3052, Australia. [4] Division of Experimental Animal Pathology, The Netherlands Cancer Institute, Plesmanlaan 121, 1066 CX Amsterdam, The Netherlands. [5] School of Life Sciences, Fudan University, Shanghai, China. [6] Department of Biochemistry and Molecular Biology and The Bio21 Molecular Science and Biotechnology Institute, The University of Melbourne, Parkville, VIC 3052, Australia. [7] School of Life Sciences, Nanjing University, Nanjing 210023, China. [8] Inflammation Division, The Walter and Eliza Hall Institute of Medical Research, Parkville, VIC 3052, Australia. [9] Department of Anatomical Pathology, St Vincent's Hospital, The University of Melbourne, Fitzroy, VIC 3065, Australia. [10] Department of Surgery, St Vincent's Hospital, The University of Melbourne, Fitzroy, VIC 3065, Australia. [11] School of Mathematics and Statistics, The University of Melbourne, Parkville, VIC 3052, Australia. Correspondence and requests for materials should be addressed to K.D.S. (email: sutherland.k@wehi.edu.au)

The Kirsten Rat Sarcoma viral oncogene homolog (KRAS) oncoprotein is a critical driver in 22% of all cancers[1]. Constitutive GTPase function is gained through a point mutation, typically at codon 12, 13 or 61[2], responsible for the hyperactivation of key proliferative and survival pathways. Unlike other key oncoproteins in cancer (i.e., EGFR, HER2, ALK), KRAS has remained an elusive clinical target in cancer due to its perceived undruggable nature[3]. In lung cancer, KRAS is most frequently mutated in lung adenocarcinoma (LUAD), where it is altered in 33% of patients[4].

Genetically engineered mouse models (GEMMs) based on temporal and spatial expression of oncogenic Kras[5,6] have proven instrumental in understanding the molecular and cellular events that underpin this genetic subset of tumors. Alveolar type 2 cells appear to be the predominant cell-of-origin of $Kras^{G12D}$-mutant tumors[6,7]. However, as seen in other lung cancer subsets, a high level of plasticity likely exists within the lung epithelium with multiple cell populations capable of transformation given an appropriate stimulus[8,9]. In murine models, co-mutation of critical tumor suppressor genes p53 and Lkb1 accelerate $Kras^{G12D}$-induced tumorigenesis and in the case of Lkb1 inactivation, alter the tumor spectrum[10,11]. In line with the model that genetic alterations can drive a distinct immune response[12], tumor-bearing lungs from $Kras^{G12D}$-mutant mice exhibit an increase in alveolar macrophages. This is thought to be driven by an inflammatory response[13], though the mechanism is not well understood. Despite this progress, chemotherapy, radiotherapy and/or surgery remain the standard-of-care for patients with KRAS-mutant lung cancer. Indeed, the recent identification of dependencies borne through common co-occurring mutations provide an appealing strategy to target KRAS-mutant lung cancer.

Approximately 20% of KRAS-mutant LUAD carry loss-of-function mutations in Kelch-like ECH-associated protein 1 (KEAP1)[14]. A key member of the antioxidant response pathway, KEAP1 functions as a negative regulator of the transcription factor nuclear factor erythroid 2-like 2 (NFE2L2/NRF2)[15]. Loss-of-function mutations in KEAP1 activate the NRF2 pathway, which in turn alters the transcription of over 200 downstream genes, involved in cellular antioxidant, detoxification, and metabolic pathways[16]. In GEMMs, we have previously described a synergy between the Keap1/Nrf2 and PI3K pathways in LUAD[17]. However, controversy exists over the capacity of Keap1 to function as a tumor suppressor in the context of $Kras^{G12D}$-driven lung cancer[18–21], where the ability to accelerate tumorigenesis in the absence of additional loss-of-function mutations in p53 or Lkb1 remains unclear[18].

Here, we identify dependencies in the KEAP1-mutant LUAD subgroup through the generation of GEMMs based on Cre-inducible expression of oncogenic $Kras^{G12D}$ and combined loss of Keap1. Strikingly, we show that Keap1-deficient $Kras^{G12D}$ tumors arise from a bronchiolar cell-of-origin, without the concomitant induction of pro-tumorigenic alveolar macrophage expansion observed in tumors originating from alveolar cells. Furthermore, Keap1 loss reprogrammed the metabolic wiring of oncogenic $Kras^{G12D}$ tumor cells by hijacking the pentose phosphate pathway (PPP). Treatment with the PPP inhibitor 6-aminonicotinamide (6-AN) abrogated the growth of Keap1-deficient tumor cells, suggesting a potential therapeutic approach to target this subset of KRAS-mutant LUAD cancers.

## Results

**KEAP1 alterations are enriched in KRAS-mutant LUAD.** To investigate the key genetic alterations in KRAS-mutant NSCLC, we interrogated large publicly available datasets from the Broad Institute[22], the Cancer Genome Atlas[4] and the Memorial Sloan Kettering Cancer Center (MSKCC)[23] using cBioPortal[24,25]. KRAS was mutated in 36.9% of cases (Fig. 1a and Supplementary Table 1), with a notable increase in mutation frequency of KEAP1 in KRAS-mutant cases compared to KRAS-WT (Fig. 1b). Top co-mutated genes TP53, KEAP1 and STK11 were interrogated for their mutual exclusivity and co-mutation frequency. In accordance with previous findings[14], TP53 and KEAP1 were seldom co-mutated, while co-mutation of KEAP1 and STK11 was more frequent (Fig. 1c and Supplementary Fig. 1 and Supplementary Table 1). Importantly, a significant proportion of KRAS-mutant LUAD were mutant for either KEAP1 or STK11, highlighting that these mutations also occur independently.

To verify these findings in an independent LUAD dataset, we mined 161 KRAS-mutant patient samples from the Clinical Lung Cancer Genome Project (CLCGP)[26]. Patient stratification by co-mutation found a similar frequency of KEAP1-only, KEAP1/ STK11, TP53 only, TP53/STK11, and STK11 only samples (Fig. 1d and Supplementary Data 1). Interestingly, KRAS-mutant tumors co-mutated with TP53 or KEAP1 only were associated with increased tumor stage (Fig. 1e, f) suggesting that inactivation of these tumor suppressors drives a more aggressive tumor phenotype. Furthermore, expression of the NRF2 transcriptional target, NAD(P)H:quinone dehydrogenase 1 (NQO1) was elevated in KRAS-mutant LUAD, co-mutant for KEAP1 (Fig. 1g), confirming its potential as a clinical biomarker for this subgroup of patients[17,18].

**Keap1 loss accelerates $Kras^{G12D}$-induced lung tumorigenesis.** To date, CRISPR/Cas9 technology has been utilized to investigate the functional consequences of Keap1 inactivation concomitant with activation of oncogenic $Kras^{G12D}$ [19,20]. Discrepancies, however, exist between these studies and the clinical relevance questionable, with models generated reflecting a minor subclass of human tumors[18]. To overcome this, we crossed loxP-STOP-loxP $Kras^{G12D/+}$ [5] (K) with mice carrying the $Keap1^{flox/flox}$ allele[27] ($Kras^{G12D/+};Keap1^{flox/flox}$; hereafter KK). Consistent with the aggressive nature of KEAP1-mutant LUAD[18,28], Ad5-CMV-Cre-infected KK mice exhibited a significantly reduced lifespan compared to K mice, with a median survival of 71 and 127 days, respectively (Fig. 2a; Mantel-Cox test $p > 0.0001$). At autopsy, moribund mice had significant lung tumor burden (Fig. 2b) and no additional abnormalities or metastases. In line with reduced survival rates, the lungs of KK mice, analyzed 3 months following Ad5-CMV-Cre infection, were significantly heavier (Fig. 2c), and displayed an increased number of lesions (Fig. 2d) compared to the lungs of K mice. Quantification of the tumor spectrum revealed a significant increase in the frequency of adenomas in KK mice compared to K only control mice (Fig. 2e). Consistent with activation of the Nrf2 pathway following loss of Keap1, Nrf2 was concentrated in the nucleus of recombined cells (Fig. 2f) in KK mice expressing a YFP reporter allele[29]. Moreover, cell lines derived from KK tumors displayed a greater clearance of reactive oxygen species (Fig. 2g), further exemplifying the enhanced function of the Nrf2 pathway in tumors with loss of Keap1. Together, these data indicate that Keap1 is a potent tumor suppressor in $Kras^{G12D}$-induced lung tumorigenesis.

While these findings suggest that Keap1 can exert its tumor suppressive function in a p53- and Lkb1-indpendent manner, the co-occurrence of KEAP1 mutations with STK11, and to a lesser extent TP53, in KRAS-mutant tumors may imply that cancer cells gain additional growth advantage. To systematically evaluate this hypothesis, $Keap1^{flox/flox}$ mice were crossed with $Kras^{G12D/+}$; $p53^{flox/flox}$ mice[10] (hereafter KPK) and $Kras^{G12D/+};Lkb1^{flox/flox}$ mice[30] (hereafter KLK). $Kras^{G12D/+};p53^{flox/flox}$ (KP) and $Kras^{G12D/+};Lkb1^{flox/flox}$ (KL) mice served as controls. Lung

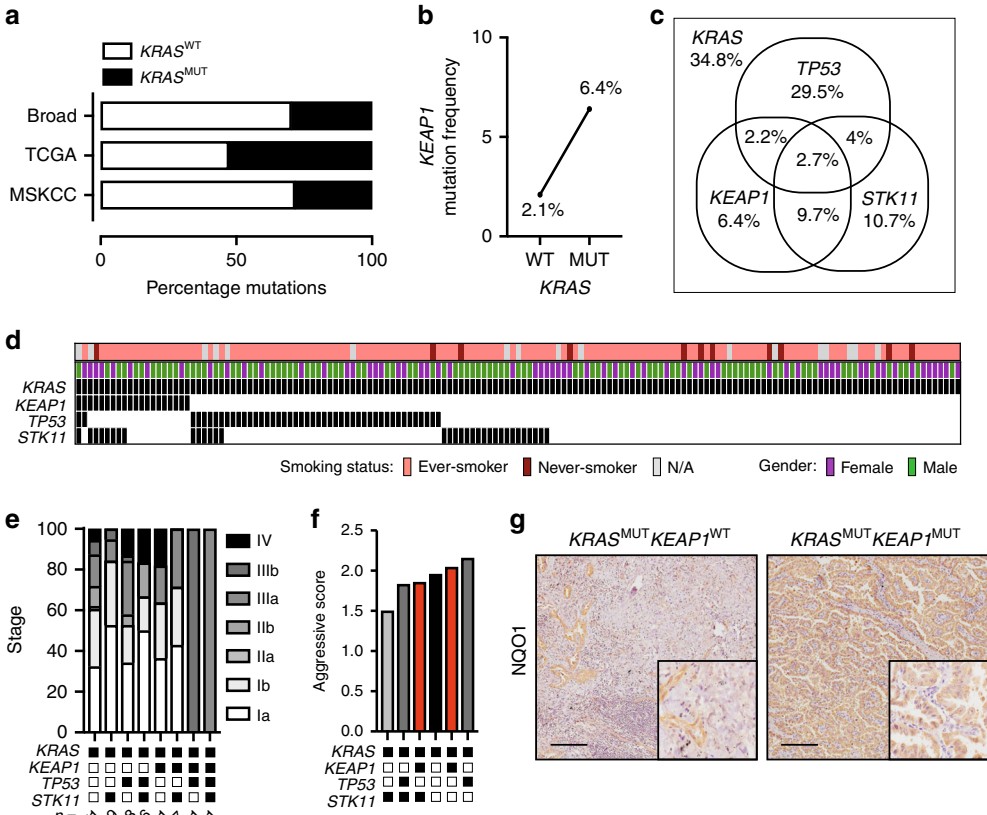

**Fig. 1** *KEAP1* mutation is enriched in *KRAS*-mutant lung adenocarcinoma. **a** Frequency of *KRAS* mutation in lung adenocarcinoma (LUAD) obtained from the Broad Institute ($n = 182$), The Cancer Genome Atlas ($n = 313$) and the Memorial Sloan Kettering Cancer Center (MSKCC) ($n = 864$). **b** Frequency of *KEAP1* mutation status in *KRAS* WT (wild type; $n = 921$) or MUT (mutant; $n = 627$) LUAD samples. **c** Venn Diagram of co-mutation/mutual exclusivity of *KEAP1*, *TP53*, and *STK11* mutations in the subset of *KRAS*-mutant LUAD ($n = 627$). **d** *KEAP1*, *TP53*, and *STK11* mutation status in *KRAS*-mutant LUAD patient samples from the Clinical Lung Cancer Genome Project (CLCGP) cohort ($n = 155$). **e** Frequency of clinical stage in subsets of *KRAS*-mutant LUAD from (**d**). **f** Aggressive score assessed by clinical stage in each *KRAS*-mutant subgroup. **g** NQO1 immunostaining on *KRAS*^MUT*KEAP1*^WT*TP53*^MUT and *KRAS*^MUT*KEAP1*^MUT*TP53*^WT patient samples. Scale, 200 μm

tumor induction was monitored in cohorts of mice following intranasal Ad5-CMV-Cre delivery. *Keap1* loss minimally impacted the survival rate of KP (KPK mice; 57 days versus KP mice; 83 days; Mantel-Cox test $p = 0.36$; Supplementary Fig. 2a) and KL (KLK mice; 49 days versus KL; 43 days) mice (Table 1). No lung tumors were observed in mice carrying bi-allelic inactivation of *Keap1* and *Lkb1* or *Keap1* and *p53* (Table 1), indicating the requirement of a collaborative oncogene to drive tumorigenesis. Consistent with activation of the Nrf2 pathway following loss of *Keap1*, lung tumor pieces from KK, KPK, and KLK GEMMs all displayed enhanced *Nqo1* transcriptional activation (Supplementary Fig. 2b), further exemplifying the enhanced function of the Nrf2 pathway in tumors with loss of *Keap1*. With the exception of KLK mice, all lesions displayed an adenomatous phenotype, with increased Nkx2.1 expression (Supplementary Fig. 2c), while the lungs of KLK mice displayed a mixture of adenomatous and squamous lesions (Supplementary Fig. 2d). Importantly, while *Keap1* does not significantly collaborate with *p53* or *Lkb1* loss to accelerate *Kras*^G12D-induced tumorigenesis, all tumor cells harboring *Keap1* inactivation exhibit augmented Nrf2 pathway activation.

**Reduced inflammatory response in *Keap1*-deficient tumors.** Given the emerging evidence linking the inactivation of specific tumor suppressors with unique immune microenvironments of *KRAS*-mutant LUAD[14,31,32], we interrogated the immune cell composition of *Keap1*-deficient tumors using a multiparametric

flow cytometry approach[17]. KK and KP mice were used for this analysis, due to similarities in tumor phenotype and latency between models (Table 1). The presence of adenomatous and squamous lesions in KLK mice (Supplementary Fig. 2d and Table 1) prohibited the use of this model, as the immune milieu of squamous tumors appears to be distinct to that of LUAD[8,33]. Consistent with previous findings[13], the alveolar macrophage compartment (CD45+CD11c+CD103−) was significantly altered in the lungs of tumor-bearing K and KP mice compared to non-tumor bearing control mice (Fig. 3a and Supplementary Fig. 3a). In stark contrast however, no expansion in alveolar macrophages was observed in the lungs of KK tumor-bearing mice. Critically, macrophage expansion was an early tumorigenic event in KP mice, with infiltration already apparent in atypical adenomatous hyperplastic (AAH) lesions (Supplementary Fig. 3b). Immuno-histochemical staining using F4/80, a well-characterized macro-phage marker, confirmed these observations in established tumors (Fig. 3b). Interestingly, no differences in the phenotype of infiltrating alveolar macrophages was detected (Supplementary Fig. 3c–e), nor were any significant alterations in the expression of *GM-CSF* or secretion of IL-6 and TNFα observed in KK and KP tumor cells (Supplementary Fig. 3f–h) that could explain the difference in macrophage recruitment between the KK and KP tumor subgroups. Intriguingly, although the latency of KK and KP mice were similar, there was an increase in carcinomatous lesions in KP mice (Fig. 3c). To investigate whether the increased macrophages were playing a role in tumor development, we

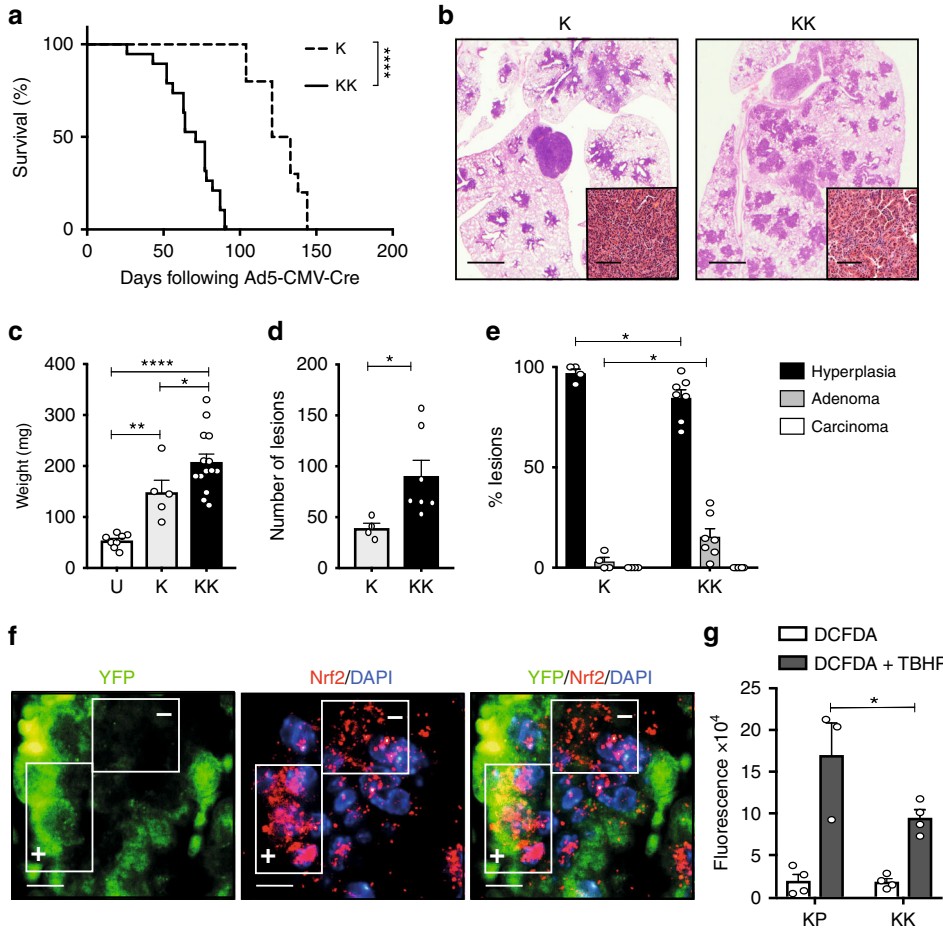

**Fig. 2** *Keap1* inactivation accelerates *Kras*[G12D]-induced lung adenocarcinoma. **a** Survival analysis of K ($n = 10$) and KK ($n = 19$) mice following Ad5-CMV-Cre administration. Mantel-Cox test: ****$p < 0.0001$. **b** Hematoxylin and eosin (H&E) stained lungs of representative K and KK mice at ethical endpoint. Scale, 1 mm; inset, 100 μm. **c** Weight of left lung lobe of uninfected littermates (U, $n = 8$), K ($n = 5$) and KK ($n = 14$) mice 3 months following Ad5-CMV-Cre infection. Ordinary one-way ANOVA/Holm-Sidak's multiple comparisons test: U v K: **$p = 0.006$; U v KK: ****$p < 0.0001$; K v KK: *$p = 0.0327$. Mean ± SEM. **d** Number of lesions in the lungs of K ($n = 4$) and KK ($n = 7$) mice 3 months post Ad5-CMV-Cre. Ordinary one-way ANOVA/Tukey's multiple comparisons test K v KK *$p = 0.0397$. Mean ± SEM. **e** Lesion classification of K ($n = 4$) and KK ($n = 7$) lungs 3 months post Ad5-CMV-Cre infection. Two-way ANOVA/Tukey's multiple comparisons test: alveolar hyperplasia K v KK *$p = 0.05$, adenoma K v KK *$p = 0.05$. Mean ± SEM. **f** Immunofluorescence of EYFP and Nrf2 expression in *EYFP*[T/+];*Kras*[G12D/+];*Keap1*[Δ/Δ] three months following Ad5-CMV-Cre infection. Boxes represent EYFP positive and negative regions. Scale, 10 μm. **g** Quantification of 2,7-dichlorofluorescin diacetate (DCFDA) alone or in combination with 110 μM tert-butyl hydrogen peroxide (TBHP) to stimulate reactive oxygen species (ROS) in primary cell lines ($n = 4$ KP and KK). Mann–Whitney test DCFDA + TBHP KP v KK *$p = 0.0286$. Mean ± SEM

### Table 1 Comparison of murine lung cancer model cohorts

| Genotype | # of mice infected | Survival (range)[a] | Histopathology |
|---|---|---|---|
| *Kras*[G12D/+] | 10 | 127 (104–144) | Adenomatous |
| *Kras*[G12D/+];*p53*[Δ/Δ] | 10 | 83 (60–91) | Adenomatous |
| *Kras*[G12D/+];*Keap1*[Δ/Δ] | 19 | 71 (26–90) | Adenomatous |
| *Keap1*[Δ/Δ17] | 26 | >365 | No lesions |
| *Keap1*[Δ/Δ];*p53*[Δ/Δ] | 10 | >365 | No lesions |
| *Kras*[G12D/+];*Lkb1*[Δ/Δ] | 8 | 43 (37–101) | Mixture of adenomatous and squamous |
| *Keap1*[Δ/Δ];*Lkb1*[Δ/Δ] | 10 | >365 | No lesions |
| *Kras*[G12D/+];*p53*[Δ/Δ];*Keap1*[Δ/Δ] | 5 | 57 (55–104) | Adenomatous |
| *Kras*[G12D/+];*Lkb1*[Δ/Δ];*Keap1*[Δ/Δ] | 9 | 49 (30–64) | Mixture of adenomatous and squamous |

[a]Median survival is represented as days following intra-nasal injection of $2 \times 10^8$ PFU Ad5-CMV-Cre virus

reduced alveolar macrophage numbers in KP mice through intranasal administration of Clodronate-loaded liposomes (Fig. 3d). Alveolar macrophages were effectively reduced in KP lungs to levels comparable to that of non-tumor bearing mice (U)

12 weeks following Ad5-CMV-Cre (Fig. 3e, f). Strikingly, the epithelial compartment in clodronate treated KP mice was significantly reduced compared to PBS control-treated mice (Fig. 3g, h). Consistent with this finding, tumor size was reduced in

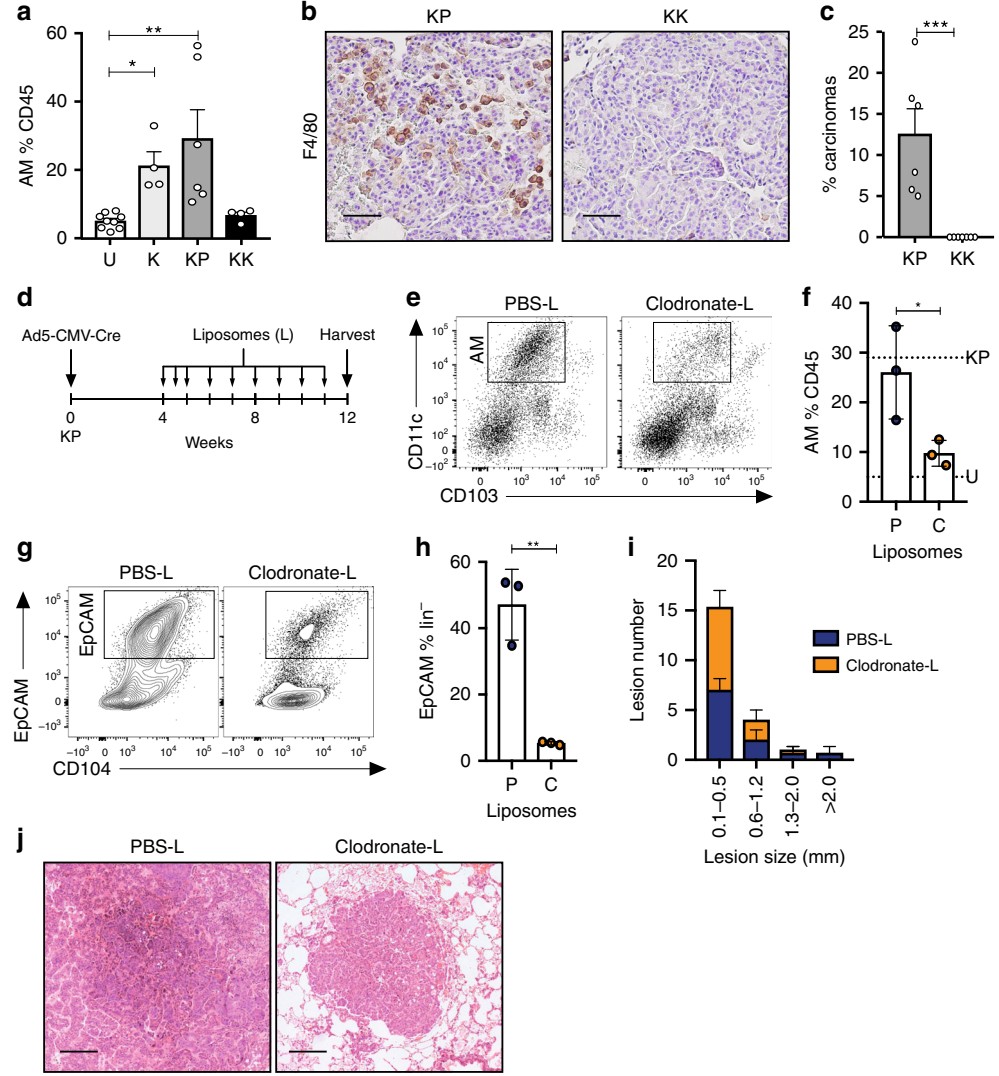

**Fig. 3** Alveolar macrophages contribute to *Kras*[G12D]-induced tumorigenesis. **a** Quantification of alveolar macrophages (CD11c[+]CD11b[−]CD103[−]) in CD45[+] immune lung infiltrate in uninfected (U; *n* = 9) and K (*n* = 4), KP (*n* = 6), KK (*n* = 4) mice 3 months post Ad5-CMV-Cre infection. Kruskal–Wallis test/ Dunn's multiple comparisons test U v K \**p* = 0.0127; U v KP \*\**p* = 0.0042. Mean ± SEM. **b** F4/80 immunostaining on KP and KK lung tissue 3 months post Ad5-CMV-Cre infection. Scale, 50 μm. **c** Frequency of carcinomatous lesions in KP (*n* = 6) and KK (*n* = 7) lungs 3 months post Ad5-CMV-Cre infection. Unpaired *t* test \*\*\**p* = 0.001. Mean ± SEM. **d** Schematic of treatment plan. Briefly, KP mice were administered weekly PBS-liposomes or Clodronate-liposomes four weeks following Ad5-CMV-Cre infection. Lungs were harvested and analyzed for alveolar macrophages and tumor burden following 8 weeks of liposome treatment. **e** Representative flow cytometry plot of alveolar macrophage population (CD45[+]CD11c[+]CD103[−]) in the lungs of PBS- or Clodronate-liposome (L) treated mice. **f** Quantification of alveolar macrophages as a proportion of CD45[+] cells in the lungs of KP mice treated with PBS (P; *n* = 3) or Clodronate (C; *n* = 3) liposomes. Dotted line represents mean value of alveolar macrophages in KP and U from (**a**), above. Unpaired *t* test \**p* = 0.0443. Mean ± SD. **g** Representative flow cytometry plot of epithelial cells (EpCAM[+]) in the lungs of PBS or Clodronate liposome treated mice. **h** Quantification of EpCAM[+] population as a proportion of lineage negative (CD31[−]CD45[−]) cells. Unpaired *t* test \*\**p* = 0.0025. Mean ± SD. **i** Quantification of lesion size in PBS- and Clodronate-liposome (L) treated mice. **j** Representative H&E of PBS- and Clodronate-liposome (L) treated mice. Scale, 200 μm

clodronate treated KP mice (Fig. 3i, j and Supplementary Fig. 3i). Taken together, these findings suggest that alveolar macrophages infiltrating the lungs of KP mice are tumor-promoting.

**Bronchiolar cells are sensitive to Nrf2 hyperactivation.** To investigate the absence of alveolar macrophage infiltration in KK tumor-bearing lungs, we dissected the mechanisms by which alveolar macrophages interact with epithelial cells. Alveolar macrophages are believed to directly interact with alveolar type 2 (AT2) epithelial cells via CD200[34,35] and/or the gap junction Connexin-43 (Cx43)[36]. Whilst no difference in *CD200* expression was detected between KP and KK tumors (Supplementary

Fig. 4a), significantly lower *Cx43* expression was observed in FACS-isolated tumor cells (Fig. 4a) and alveolar macrophages (Fig. 4b) from KK lungs. To evaluate this relationship in patient samples, we curated a consensus NRF2 signature based on published NRF2 signatures (Supplementary Fig. 4b) and stratified *KRAS*-mutant LUAD samples from the TCGA cohort into low and high NRF2 subgroups based on NRF2 signature expression (Supplementary Fig. 4c). Critically, *GJA1/CX43* expression was significantly decreased in *KRAS*-mutant LUAD with a high NRF2 score (Supplementary Fig. 4d), suggesting that NRF2 pathway activity negatively correlates with *Cx43* expression, concordant with the GEMM findings (Fig. 4a). In line with the

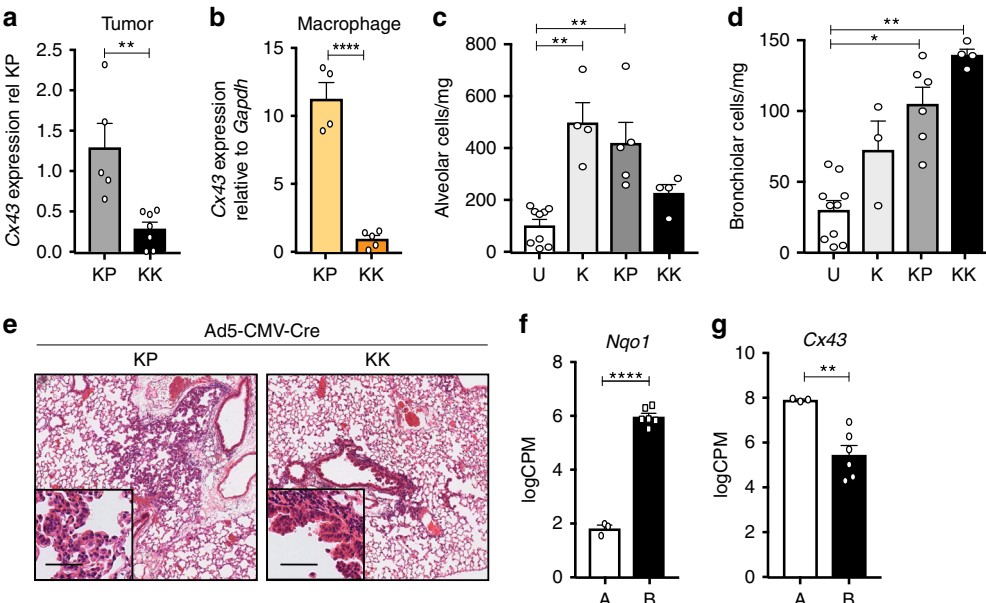

**Fig. 4** *Keap1* loss preferentially transforms bronchiolar cells. **a** Expression of *Cx43* in KP ($n = 5$) and KK ($n = 7$) tumor pieces, relative to KP. Mann–Whitney test **$p = 0.0025$. Mean ± SEM. **b** Expression of *Connexin-43* (*Cx43*) in alveolar macrophages isolated from KP ($n = 4$) and KK ($n = 5$) tumor-bearing lungs, relative to *Gapdh* control. Unpaired student *t* test ****$p < 0.0001$. Mean ± SEM. **c** Quantification of alveolar (EpCAM+CD104−) compartment in uninfected (U; $n = 9$) and K ($n = 4$), KP ($n = 4$) and KK ($n = 5$) 3 months post Ad5-CMV-Cre administration. Kruskal–Wallis test/Dunn's multiple comparisons test, U v K **$p = 0.0034$, U v KP **$p = 0.0095$. Mean ± SEM. **d** Quantification of bronchiolar (EpCAM+CD104+) compartment in uninfected (U; $n = 9$) and K ($n = 3$), KP ($n = 4$) and KK ($n = 6$) 3 months post Ad5-CMV-Cre administration. Kruskal–Wallis test/Dunn's multiple comparisons test, U v KK **$p = 0.0014$, U v KP *$p = 0.0308$. Mean ± SEM. **e** Representative H&E stained KP and KK lungs 3 weeks post Ad5-CMV-Cre infection. Scale, 200 μm; inset, 50 μm. **f** Expression of *Nqo1* in alveolar (A) and bronchiolar (B) cells ($n = 3$). ****FDR $p < 0.0001$. Mean ± SEM. **g** Expression of *Cx43* mRNA in alveolar (A) and bronchiolar (B) cells ($n = 3$). **FDR $p = 0.0055$. Mean ± SEM

utilization of *NQO1* as a clinical biomarker of NRF2 pathway activity, expression of *NQO1* alone stratified the NRF2 signature (Supplementary Fig. 4e) and distribution of *KRAS*-mutant LUAD samples from the TCGA cohort based on *NQO1* expression similarly negatively correlated with *GJA1*/*CX43* expression (Supplementary Fig. 4f). Importantly, these findings further support the use of NQO1 as a single-gene biomarker for NRF2 pathway activity. Next, we sought to functionally assess the importance of Cx43 gap junctions in the interaction between alveolar cells and alveolar macrophages. We exposed the lungs of healthy adult mice to intranasal inhalation of an intracellular inhibitor of Cx43, Gap19 peptide. Extended exposure to Gap19 resulted in the displacement of alveolar macrophages from the lungs (Supplementary Fig. 4g), confirming the importance of Cx43 in mediating epithelial-immune cell interactions. Consistent with the proposed expression of Cx43 in alveolar cells themselves[36], the tumor-bearing lungs of KK mice displayed a reduced alveolar compartment (Fig. 4c). Interestingly, and consistent with a bronchiolar trophism observed in previous models of *Keap1* loss[17], KK tumor-bearing lungs were characterized by elevated levels of EpCAM+CD104+ bronchiolar cells (Fig. 4d) and increased expression of *Scgb1a1*, a marker of Club cells (Supplementary Fig. 5a). To further evaluate the cell-types that undergo initial transformation upon recombination, lungs of KP and KK mice were analyzed 3-weeks following Ad5-CMV-Cre infection, prior to the onset of overt malignant disease (Supplementary Fig. 5b). AAH was pronounced in the lungs of KP mice (Fig. 4e and Supplementary Fig. 5c), consistent with an alveolar cell-of-origin in *Kras*G12D induced tumors[6,7,37]. In contrast, hyperplasia of the bronchiolar epithelium was predominant in the lungs of KK mice at this time, while AAH was not detected (Fig. 4e and Supplementary Fig. 5d). To interrogate why bronchiolar cells are more sensitive to transformation upon

enhanced activation of the Nrf2 pathway, we performed KEGG pathway analysis of alveolar and bronchiolar cells purified from normal murine lung epithelium[38]. Although primary transcript levels of *Nrf2* and *Keap1* were comparable between the two epithelial compartments (Supplementary Fig. 6a), ubiquitin ligase pathways that mediate Nrf2 post-translational degradation were expressed at significantly higher levels in alveolar cells (Supplementary Fig. 6b; Differential ROAST analysis $p = 1.29 \times 10^{-9}$). In line with these findings, expression of antioxidant genes and glutathione transferases regulated by the Nrf2 pathway were enriched in the bronchiolar compartment (Supplementary Fig. 6c; Differential ROAST analysis $p = 5 \times 10^{-4}$) as was the expression of Nrf2 target gene *Nqo1* (Fig. 4f). To further dissect this relationship, we applied the NRF2 consensus signature score (Supplementary Fig. 4b) to differentially expressed genes in the alveolar and bronchiolar compartments. Interestingly, a significant correlation was observed between the NRF2 signature and the bronchiolar compartment (Supplementary Fig. 6d). Together, these findings suggest that bronchiolar cells have increased basal activity of the Nrf2 pathway and thus are more responsive to Nrf2-mediated transformation. Moreover, consistent with previous imaging observations[36], *Cx43* expression was enriched specifically in normal alveolar epithelium (Fig. 4g). Taken together, these findings suggest that the decreased expression of *Cx43* (Fig. 4a) and absence of macrophage expansion (Fig. 3a) may be due to a distinct bronchiolar, rather than alveolar, cell-of-origin of KK tumors.

**The cell-of-origin dictates the immune microenvironment.** To further explore whether the phenotype of the cancer-initiating cells directly influences the immune contexture, cell-type specific Ad5-Cre viruses were used to initiate tumors in KK and KP

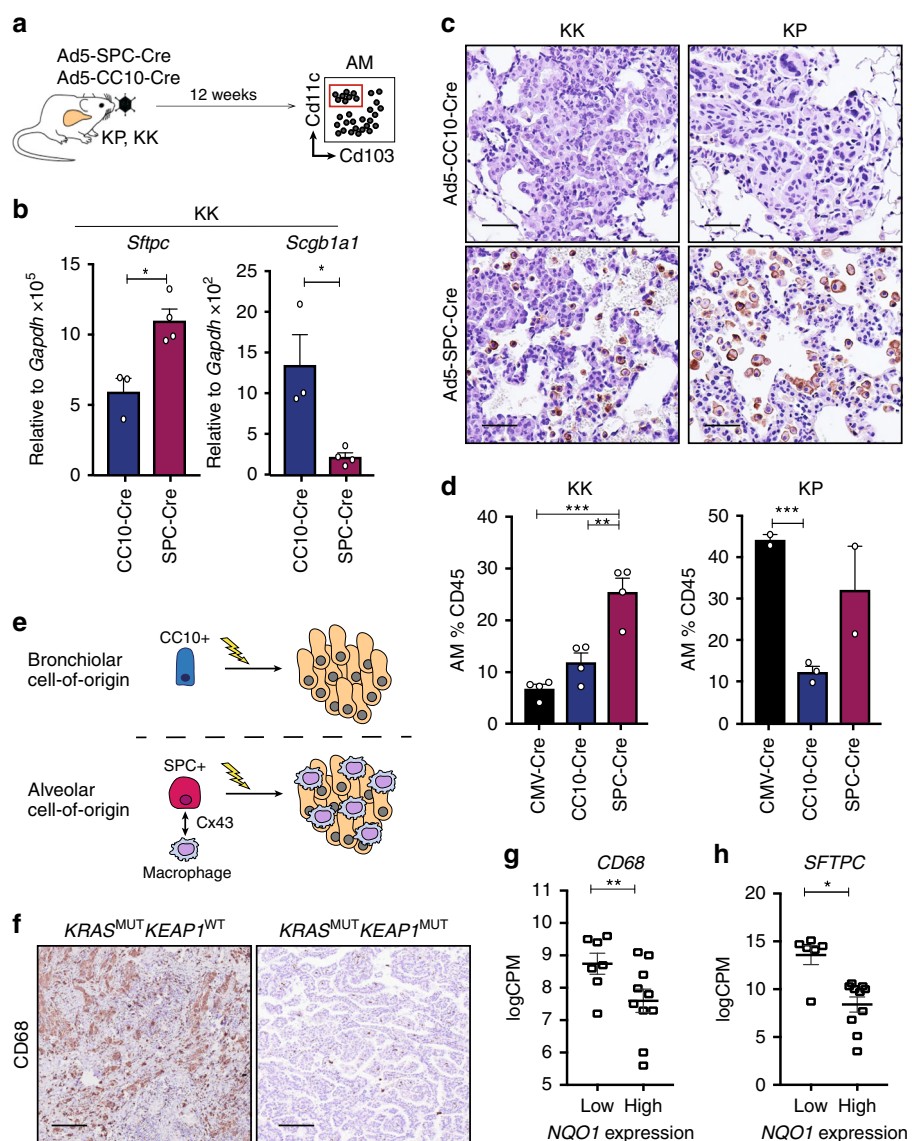

**Fig. 5** Cell-of-origin determines the alveolar macrophage composition in LUAD. **a** Schematic of cell type-specific Ad5-Cre experimental design. **b** Expression of *Sftpc* and *Scgb1a1* in KK lung tumor pieces 3 months following administration of Ad5-CC10-Cre ($n = 3$) or Ad5-SPC-Cre ($n = 4$). Unpaired *t* test, *Sftpc* *$p = 0.0168$; *Scgb1a1* *$p = 0.0107$. Mean ± SEM. **c** F4/80 immunostaining on lung tissue from KK and KP mice infected with Ad5-CC10-Cre or Ad5-SPC-Cre. Scale, 50 μm. **d** Quantification of alveolar macrophages in CD45+ immune lung infiltrate of Ad5-CMV-Cre infected KK ($n = 4$) and KP ($n = 4$), Ad5-CC10-Cre infected KK ($n = 4$) and KP ($n = 2$) and Ad5-SPC-Cre infected KK ($n = 4$) and KP ($n = 3$) mice. KK, ordinary one-way ANOVA/Holm-Sidak's multiple comparisons test CMV-Cre v SPC-Cre ***$p = 0.0002$, SPC-Cre v CC10-Cre **$p = 0.0016$. KP, unpaired *t* test CMV-Cre v CC10-Cre ***$p = 0.0006$. Mean ± SEM. **e** Schematic of cell-of-origin hypothesis and enhanced macrophage number in lung tumors arising from alveolar epithelial cells. Lightning bolt depicts genetic alterations. **f** Immunostaining of CD68 on KRAS$^{MUT}$KEAP1$^{WT}$ and KRAS$^{MUT}$KEAP1$^{MUT}$ LUAD. Scale, 200 μm. **g** Analysis of *CD68* expression in NQO1$^{low}$ ($n = 7$) and NQO1$^{high}$ ($n = 10$) KRAS-mutant LUAD TCGA patient samples. **FDR $p = 0.0012$. Mean ± SEM. **h** Analysis of *SFTPC* expression in NQO1$^{low}$ ($n = 7$) and NQO1$^{high}$ ($n = 10$) KRAS-mutant LUAD TCGA patient samples. **FDR $p = 0.0108$. Mean ± SEM

mice specifically in AT2 (Ad5-SPC-Cre) or bronchiolar Club (Ad5-CC10-Cre) cells in the lung[7,9] (Fig. 5a). Consistent with the cell restricted nature of the viruses, *Sftpc* (pro-SPC) expression was significantly enhanced in tumors initiated from AT2 cells, while elevated expression of *Scgb1a1* (CC10) was detected in tumors following Ad5-CC10-Cre infection (Fig. 5b and Supplementary Fig. 7a). Strikingly, F4/80+ infiltrating macrophages were only observed in KP and KK tumors initiated from an AT2 cell-of-origin (Fig. 5c). This restricted expansion of the macrophage compartment was further validated by flow cytometric analysis of alveolar macrophages (Fig. 5d and Supplementary Fig. 7b). Together, these findings suggest that the cell-of-origin, rather than the genetic alteration,

is the commanding determinant of alveolar macrophage expansion in *KRAS*-mutant LUAD (Fig. 5e).

To determine whether the effect of NRF2 pathway activation on the immune microenvironment is conserved in patient cohorts, we next curated an independent LUAD cohort of FFPE blocks with known *KRAS* and *TP53* mutation status ($n = 48$), hereafter the St Vincent's cohort (Supplementary Data 2). Immunostaining of NQO1 and CD68 revealed a negative correlation between macrophage infiltration and NRF2 pathway activity (Supplementary Fig. 7c, d), concordant with the GEMM findings. Furthermore, patient samples of known *KEAP1* mutation status were evaluated for CD68 intensity, revealing reduced macrophage infiltration in *KRAS*$^{MUT}$*KEAP1*$^{MUT}$

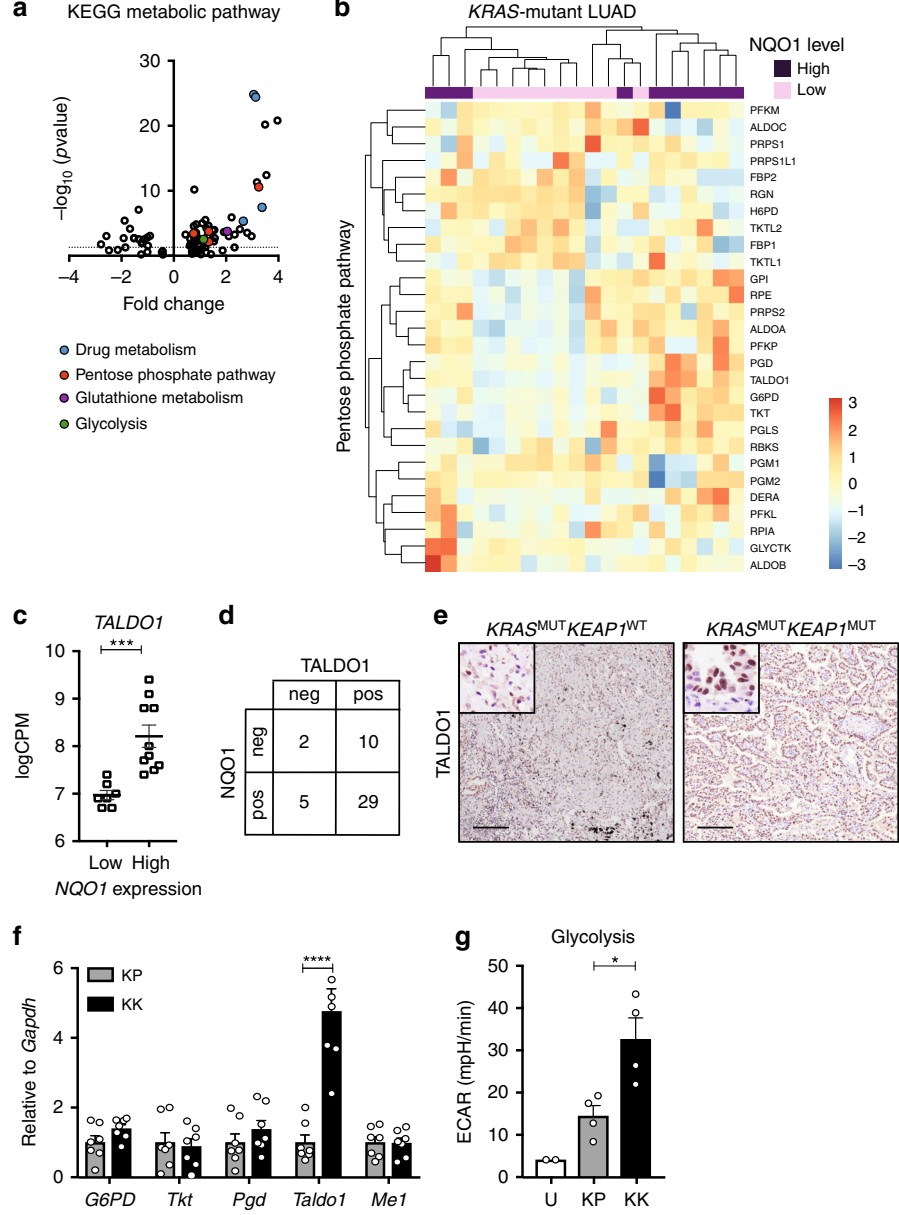

**Fig. 6** Enhanced glucose metabolism in *Keap1*-mutant *Kras*[G12D] LUAD. **a** Volcano plot of top KEGG metabolic pathways differentially expressed in *NQO1*[low] (*n* = 10) and *NQO1*[high] (*n* = 10) *KRAS*-mutant LUAD from the TCGA dataset. Pathways involved in drug metabolism, pentose phosphate pathway, glutathione metabolism, and glycolysis have been highlighted. Dotted line represents *p* = 0.05. **b** Heatmap of KEGG pentose phosphate pathway in *NQO1*[low] (*n* = 10) and *NQO1*[high] (*n* = 10) *KRAS*-mutant LUAD from the TCGA dataset. **c** Analysis of *TALDO1* expression in *NQO1*[low] (*n* = 7) and *NQO1*[high] (*n* = 10) *KRAS*-mutant LUAD TCGA patient samples. ***FDR *p* = 0.0006. Mean ± SEM. **d** Correlation of TALDO1 and NQO1 immunostaining in *KRAS*-mutant LUAD (*n* = 46). **e** TALDO1 immunostaining on *KRAS*[MUT]*KEAP1*[WT] and *KRAS*[MUT]*KEAP1*[MUT] patient samples. Scale, 200 μm; inset, 50 μm. **f** Expression of glycolytic enzyme (*G6PD*), pentose phosphate pathway enzymes (*Tkt, Pgd, Taldo1*) and malic enzyme (*Me1*) in tumor pieces from KP (*n* = 7) and KK (*n* = 7) lungs. Expression relative to *Gapdh* housekeeper control and quantified relative to KP expression. Two-way ANOVA/Sidak's multiple comparisons test *Taldo1* KP v KK ****p* < 0.0001. Mean ± SEM. **g** Rate of glycolysis measured by extracellular acidification rate (ECAR) of uninfected lung epithelium (U; *n* = 2 experiments, *n* = 6 mice per experiment) and KP (*n* = 4) as well as KK (*n* = 4) flow cytometry isolated tumor cells 3 months following Ad5-CMV-Cre infection. Ordinary one-way ANOVA/Holm-Sidak's multiple comparisons test, KP v KK **p* = 0.0198. Mean ± SEM

patient samples (Fig. 5f). In the independent TCGA LUAD cohort, *CD68* expression was significantly reduced in the *KRAS*[MUT]*NQO1*[high] patient samples (Fig. 5g; FDR *p* = 0.0012), as was the expression of *SFTPC*, a specific marker of AT2 cells (Fig. 5h; FDR *p* = 0.0108). Thus, loss-of-function mutations in *KEAP1* alter the immune microenvironment by facilitating malignant transformation from bronchiolar epithelial cells.

**TALDO1 is upregulated in LUAD with high NRF2 activity.** Having established the unique nature of the bronchiolar-induced immune microenvironment in *KEAP1*-mutant *KRAS*-driven LUAD, we sought to identify the underlying mechanisms of tumor progression in this genetic subtype. We surveyed KEGG pathways differentially expressed in the *KRAS*[MUT]*NQO1*[high] versus *KRAS*[MUT]*NQO1*[low] TCGA cohort, and identified upregulation of pathways including the pentose phosphate pathway

(PPP), glutathione metabolism and drug metabolism (Fig. 6a), consistent with increased NRF2 pathway activity[15,18,39]. Both the PPP as a whole (Fig. 6b) and the specific expression of PPP enzyme *TALDO1* (Fig. 6c) significantly correlated with the *KRAS*[MUT]*NQO1*[high] TCGA dataset. In addition, TALDO1 protein expression significantly correlated with NQO1 expression in the independent St Vincent's cohort (Fig. 6d and Supplementary Data 2). Importantly, *KRAS*[MUT]*KEAP1*[MUT] LUADs were strongly positive for TALDO1 protein expression, while *KRAS*[MUT]*KEAP1*[WT] LUADs were not (Fig. 6e). This correlation was conserved in the GEMMs, whereby *Taldo1* mRNA and protein expression were substantially increased in KK tumors, in comparison to levels in KP tumors (Fig. 6f and Supplementary Fig. 8a). Functionally, we tested the glycolytic rate in tumor cells from KK and KP mice using a protocol developed specifically for freshly isolated primary cells (Supplementary Fig. 8b). We found that KK tumor cells exhibited both an increased basal acidification rate and overall glycolytic rate compared to KP tumors (Fig. 6g and Supplementary Fig. 8c, d). Together, these findings highlight that loss of *Keap1* rewires cellular metabolism by processes distinct to that of *p53* to potentiate *Kras*[G12D]-induced tumorigenesis

***KEAP1*-mutant LUAD are sensitive to 6-AN treatment**. To dissect whether increased PPP activity results in a metabolic vulnerability in *KEAP1*-mutant tumors, we used 6-AN, an inhibitor of 6-phosphogluconate dehydrogenase (PGD)[40,41], integral to both the oxidative and non-oxidative arms of the PPP (Supplementary Fig. 9a). 6-AN significantly reduced colony forming potential in *Keap1*-deficient primary cell lines derived from KPK and KK tumors, and had no effect on the growth of KP tumor cells (Supplementary Fig. 9b, c). To investigate the therapeutic potential of targeting PPP in *Keap1* mutant tumors in a spontaneous lung tumor setting, 40 days post Ad5-CMV-Cre administration, KK and KP mice were treated with three cycles of 6-AN (Fig. 7a). While extensive adenomas formed in vehicle-treated KK mice, only hyperplasia was observed in 6-AN-treated KK mice (Fig. 7b). Consistent with histological examination, there was a significant reduction in lung weight of 6-AN treated KK mice (Fig. 7c), and an increase in unobstructed airway, quantified as a decreased hyperplasia to airway ratio (Fig. 7d). Conversely, KP mice treated with 6-AN exhibited no reduction in adenomatous lesion formation (Supplementary Fig. 9d) nor change in lung weight (Supplementary Fig. 9e) or hyperplasia parameters (Supplementary Fig. 9f, g), relative to vehicle control. Together, these findings highlight a specific vulnerability of *Keap1*-mutant tumors to inhibition of PPP activity in vivo.

To explore the sensitivity of human NSCLC cell lines to 6-AN, we used *KRAS*-mutant A549, H460 (*TP53*[WT];*STK11*[MUT];*KEAP1*[MUT]), H358 and H441 (*TP53*[MUT];*STK11*[WT];*KEAP1*[WT]). *KEAP1*-mutant cell lines displayed increased NQO1 and TALDO1 expression (Supplementary Fig. 10), more aggressive behavior (Supplementary Fig. 11a) and greater ROS clearance capacity (Supplementary Fig. 11b), indicative of enhanced NRF2 pathway activity. In this setting, the activity of 6-AN was confirmed to impact the glycolytic, and not oxidative, capacity of *KEAP1*-mutant cell lines (Supplementary Fig. 11c), consistent with inhibition of PGD[42]. In line with mouse cell line and spontaneous tumor studies, *KEAP1*-mutant cell lines displayed increased sensitivity to PPP inhibition (IC50 *KEAP1*[MUT] 9.9 ± 6.4 μM vs. *KEAP1*[WT] 44.5 ± 7.9 μM; Supplementary Fig. 10d). Furthermore, the sensitivity was more rapid in *KEAP1*-mutant cell lines (Supplementary Fig. 10e) and abrogated cell growth and survival with increased expediency (Fig. 7e and Supplementary Fig. 10f). Moreover, to examine the in vivo response of the human cell lines

to 6-AN, a subcutaneous xenograft transplantation of A549 and H358 was performed in immune-deficient mice. Consistent with the spontaneous GEMM treatment study, A549 recipient mice treated with 6-AN displayed reduced tumor growth (Fig. 7f, g) and increased survival (Fig. 7h), while H358 recipient mice received no survival benefit from 6-AN (Fig. 7i, j). Together, these findings suggest a specific dependency of *KRAS*[MUT]*KEAP1*[MUT] tumors to the PPP, that can be abrogated in vivo using pharmacological inhibitors of the pathway.

## Discussion

In this study, we directly addressed the heterogeneity of *KRAS*-mutant LUAD through the generation of a series of GEMMs. We demonstrate that *KEAP1* is a potent tumor suppressor that promotes malignancy through a metabolic dependency on the PPP. Together with our previous studies[17], our data indicate that bronchiolar epithelium are particularly sensitive to *Keap1* inactivation, with tumor initiation from this cellular compartment imposing a profound effect on the immune microenvironment of *KRAS*-mutant LUAD. We reveal a pro-tumorigenic function for macrophages in *Kras*[G12D] lung tumors induced from alveolar cells. Together, these data highlight that *KRAS*-mutant cancers hijack diverse processes to evolve. Importantly, these differing mechanisms should be considered therapeutically in order to effectively impede tumor progression.

Although KRAS can activate a basal level of NRF2 activity[43,44], the enrichment of *KEAP1* alterations in *KRAS*-mutant LUADs suggest that augmented NRF2 pathway activation is favorable in driving malignant progression. Indeed, utilizing *Cre-loxP* recombination technology, we demonstrate that *Keap1* inactivation accelerates *Kras*[G12D]-induced tumorigenesis. Interestingly, CRIPSR/Cas9 genomic editing of *Keap1* failed to accelerate *Kras*[G12D]-induced tumorigenesis in an in vivo boutique screen[19,20], while transduction of an independent sg*Keap1* lentiviral construct led to accelerated tumorigenesis on the KP background[18]. These findings suggest that, while CRISPR models may mimic the heterogeneous mutation landscape of patient tumors, in the context of defining tumor suppressors, consistent bi-allelic inactivation using *Cre-loxP* models provide a definitive genetic tool. Interestingly, conditional activation of Keap1 (*Keap1*[R554Q]) was recently shown to delay lung tumorigenesis on the KP background[21], highlighting the need to dissect differences between loss-of-function versus mutant proteins in a tissue- and cell-specific context, akin to the controversies surrounding *TP53*[45]. In this study, the absence of synergistic activity of *Keap1* and *p53* loss in the context of oncogenic *Kras*[G12D] activation aligns with the mutual exclusivity of these genetic alterations in *KRAS*-mutant LUAD[14]. Moreover, these findings suggest that *p53* and *Keap1* may be acting in similar tumor suppressive pathways, with co-mutated cells exhibiting no selective advantage. One might therefore speculate that targeting intact TP53, using small molecular inhibitors such as Nutlin, might be efficacious in this *TP53*[WT] subgroup of tumors.

While *KEAP1* and *STK11* are co-mutated in a subset of *KRAS*-mutant LUAD, KL and KLK mice exhibited similar survival rates following Ad5-CMV-Cre administration. Moreover, the tumor plasticity observed in *Kras*[G12D] models carrying *Lkb1* loss was also retained following combined loss of *Keap1*. The apparent dominance of SqCC tumors in KLK mice is concordant with the observed increase in *Nqo1* expression and accompanying reduction in ROS levels in SqCC tumors detected in KL mice[46], suggesting that augmentation of KEAP1-NRF2 pathway modulates tumor cell plasticity. Recent clinical evidence highlights a crucial role of *STK11* in dictating the therapeutic response to immune-based therapies[47]. This highlights the importance of

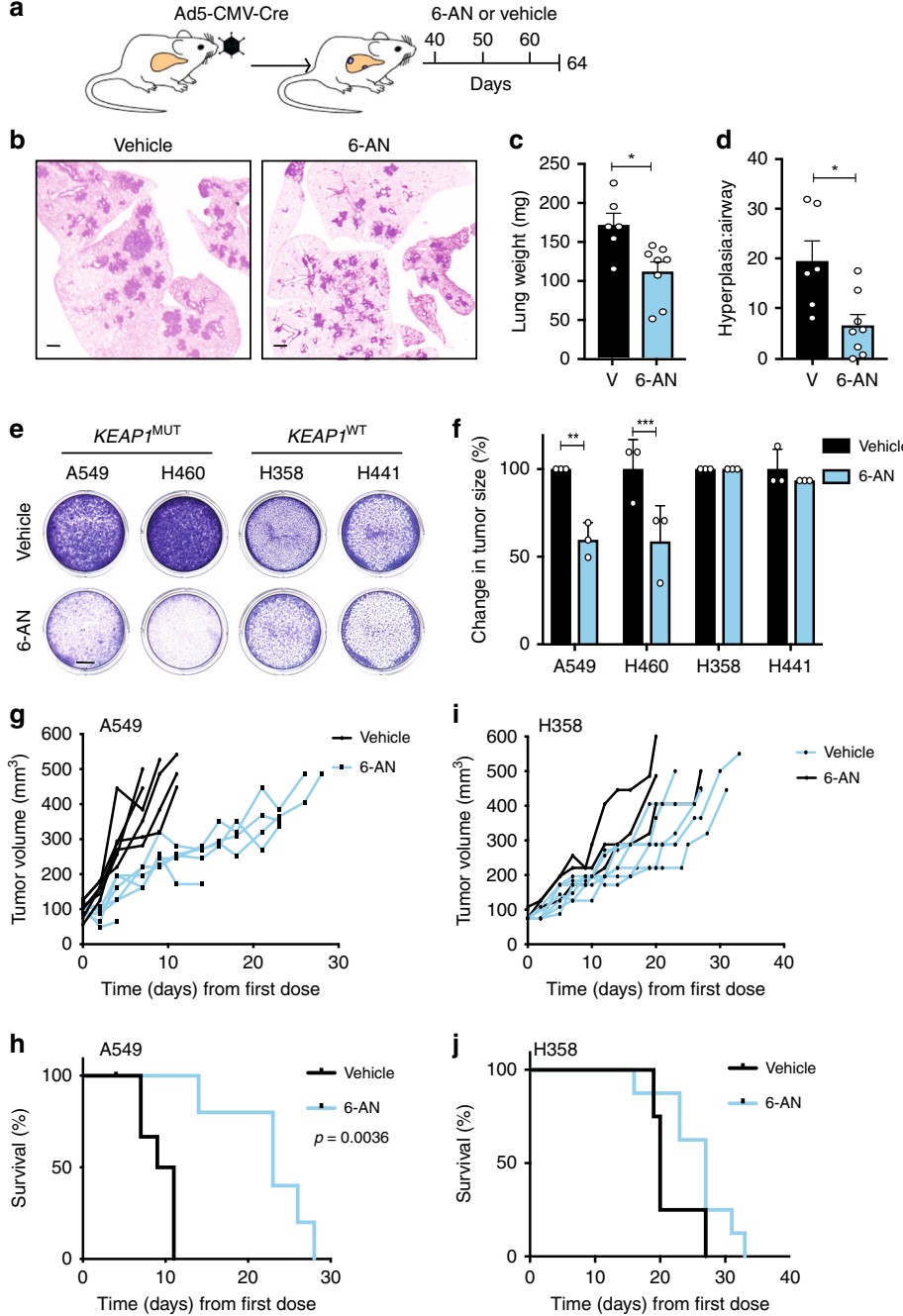

**Fig. 7** PPP blockade abrogates the growth of *KEAP1*-mutant LUAD. **a** Schematic of spontaneous tumor treatment study. Briefly, KK or KP mice were randomized into 20 mg/kg 6-AN or vehicle treated 40 days post Ad5-CMV-Cre intranasal infection. Lungs were harvested for analysis on day 64. **b** Representative H&E images of KK vehicle or 6-AN-treated lungs. Scale, 1 mm. **c** Quantification of KK superior lobe lung weight (milligrams, mg) following vehicle (*n* = 6) or 6-AN (*n* = 8) treatment. Unpaired *t* test *\*p* = 0.0106. Mean ± SEM. **d** Ratio of KK hyperplasia relative to large airway size in vehicle (*n* = 6) and 6-AN (*n* = 8) treated lungs. Unpaired *t* test *\*p* = 0.0116. Mean ± SEM. **e** Colony assay 72 h following treatment with 62.5 μM 6-AN vs vehicle in *KEAP1*^MUT and *KEAP1*^WT NSCLC cell lines. Scale, 5 mm. **f** Relative change in tumor size of xenograft models (*n* = 3/cell line) of *KEAP1*^MUT and *KEAP1*^WT 48 h following treatment with 20 mg/kg 6-AN or vehicle. Change in size relative to vehicle of each cell line. Two-way ANOVA/Sidak's multiple comparisons test: A549 **\*\*p* = 0.0012; H460 **\*\*\*p* = 0.001. Mean ± SD. **g** Tumor measurements of A549 cell line xenograft following treatment with 20 mg/kg 6-AN (*n* = 6) or vehicle (*n* = 6) every 10 days. Survival log-rank (Mantel-Cox) test **\*\*p* = 0.0023. **h** Kaplan–Meier survival curve of A549 xenograft NSG mice treated with 20 mg/kg 6-AN (*n* = 6) or vehicle (*n* = 6) every 10 days. Mantel–Cox test **\*\*p* = 0.0017. **i** Tumor measurements of H358 xenograft following treatment with 20 mg/kg 6-AN (*n* = 8) or vehicle (*n* = 4) every 10 days. **j** Kaplan–Meier survival curve of H358 xenograft NSG mice treated with 20 mg/kg 6-AN (*n* = 8) or vehicle (*n* = 4) every 10 days

establishing the contribution alteration of *KEAP1* plays in this setting, and in establishing a pure LUAD immunocompetent murine model will aid in addressing this clinically-relevant question.

The advent of immunotherapeutic agents, with in an increasing number of mechanisms to unleash immune cells to identify and kill tumor cells, has resurrected the importance of characterizing the immune microenvironment of tumors. In fact, *Sox2*-driven

models of squamous carcinomas are characterized by increased neutrophil recruitment[8,33], just as KL adenosquamous tumors displayed increased tumor associated neutrophils, while adenocarcinoma lesions did not[48]. These findings suggest that the lesion type, and corresponding cell-of-origin, may play a role in the immune cells recruited to lung tumors. While there is strong evidence that $Kras^{G12D}$-driven tumors arise from an alveolar cell-of-origin, the direct impact the initiating cell has on the well-characterized macrophage inflammation of these tumors is poorly understood[13]. We identify that restricted activation of $Kras^{G12D}$ concomitant with inactivation of either $p53$ or $Keap1$ in SPC-expressing AT2 cells resulted in the formation of lung tumors characterized by macrophage infiltration. Importantly, this data represents a paradigm shift, whereby the immune micro-environment of the tumor is directly influenced by the initiating cell, and not the genetic alterations themselves. Indeed, our analysis revealed that untransformed alveolar cells express high levels of the gap junction protein Cx43, a crucial component of the interaction between alveolar cells and alveolar macrophages. Moreover, macrophages infiltrating KRAS-mutant tumors are pro-tumorigenic, and could represent a therapeutic target in inflamed LUAD. Therapies targeting macrophages have been investigated for solid tumors, with a number of clinical trials ongoing investigating the benefits of macrophage depletion (reviewed in Poh et al.[49]). These findings highlight an immune-based approach to abrogate the growth and progression of KRAS-mutant LUAD.

The observed lack of macrophage infiltration with pro-tumorigeneic characteristics in KEAP1-mutant tumors may suggest that KEAP1-deficient tumors hijack alternative tumor-promoting mechanisms to drive and sustain tumor cell growth. Indeed, there is strong evidence linking NRF2 pathway activation and cellular metabolism[50], with recent preclinical data suggesting that inhibition of such dependencies may be a viable therapeutic approach. Here, we demonstrate the metabolic dependency of KEAP1-mutant tumors to the pentose phosphate pathway (PPP). The studies herein describe the dependency of KEAP1-mutant tumors on the induction of TALDO1, a key enzyme in the non-oxidative arm of the PPP. We and others have previously described the presence of biomarkers[17] and genetic regulation of enzyme expression[39,51] implicating the importance of PPP metabolism in KEAP1-deficient cells. Notably, the use of 6-AN may have two-fold implications on KEAP1-mutant tumors. As a PGD inhibitor, 6-AN constricts the ability of tumor cells to utilize the PPP[40] and may have additional on-target effects on glutathione (GSH) synthesis, via glutamine-cysteine ligase (GCLC)[52]. Production of GSH is an important product of NRF2 activation, essential for redox homeostasis, and therefore may provide an additional anti-tumor mechanism in 6-AN treated cells. Importantly, these findings highlight that metabolic rewiring by the NRF2 pathway is a key effector of tumor progression. Targeting metabolic pathways, either the PPP, serine biosynthesis[53,54] or glutaminase[18,55], can sufficiently block the metabolic advantage of KEAP1-mutant LUAD. These studies highlight that while robustly inhibiting tumor potential, like most other therapeutics, metabolic inhibitors will need to be applied in combination to exert a profound effect.

In conclusion, we have generated series of mouse models that recapitulate the heterogeneity of KRAS-mutant LUAD. Importantly, the identification of neoplastic abnormalities co-occurring with oncogenic KRAS must be considered in the both research models and in the clinic. Indeed, this is evidenced by the resistance to anti-PD-1 checkpoint inhibitors in KRAS-mutant LUADs harboring STK11 alterations. Based on findings presented herein, it is tempting to speculate that both the genetic alterations and cell-of-origin impact the behavior and treatment response of the tumor. Concordant with this, we demonstrate that KEAP1-deficient tumors exhibit unique characteristics dictated by their cellular origin and metabolic program. Importantly, the findings within highlight the power of targeting metabolic and immune dependencies in KRAS-mutant LUAD, thus providing an exciting alternative solution to targeting the oncoprotein itself.

## Methods

**Ethics and human samples.** We obtained all patient material according to protocols approved by the Human Research Ethics Committee of the Walter and Eliza Hall Institute of Medical Research (WEHI) and St Vincent's Hospital Human Research Ethics Committee (#10/04 and #030/12, respectively). The KRAS-mutant LUAD dataset and samples were obtained through the Clinical Lung Cancer Genome Project (CLCGP)[26] and the Victorian Thoracic Malignancies Prospective Cohort Study. Sections of KRAS-mutant LUAD patient FFPE blocks were obtained from St Vincent's Hospital (Melbourne, Australia).

**Ethics and mice.** We conducted all animal experiments according to the regulatory standards approved by the Walter and Eliza Hall Institute Animal Ethics Committee (AEC 2016.024). $Keap1^{flox}$ mice[27] were a generous gift from S. Biswal (John Hopkins Bloomberg School of Public Health). $Kras^{G12D/+}$;$p53^{flox/flox}$ mice, $Lkb1^{flox/flox}$ mice and $EYFP^{T/+}$ mice were previously described[10,29,30] and were obtained from Jackson Laboratory. All animals were maintained on a C57BL/6 background and equal proportions of males and females were used in all experiments (genotyping primers listed in Supplementary Table 2).

Seven-to-eight-week old mice were intranasally (i.n.) infected with 20 μL of $1\times10^{10}$ PFU/mL Ad5-CMV-Cre, Ad5-SPC-Cre or Ad5-CC10-Cre virus (University of Iowa Gene Transfer Core Facility #VVC-U of Iowa-5, VVC-Berns-1168) according to standard procedures[56]. At 3-weeks, 2-months or 3-months following infection, or at ethical endpoint, lungs were harvested for further analysis. Mice were injected intra-peritoneally (i.p.) 1 h prior to tissue collection with 5-Bromo-2-deoxyuridine (BrdU; 0.5 mg/10 g body weight; Sigma-Aldrich B5002).

**Flow cytometry.** The superior lobe of the left lung was dissected at the bifurcation of the left primary bronchus and weighed. Single cell suspensions were generated as described in detail previously[56]. Primary antibodies (Supplementary Table 3) were incubated for 30 min at 4 °C. Population definitions: Alveolar cells (PI⁻CD45⁻CD31⁻EpCAM⁺CD104⁻), Bronchiolar cells (PI⁻CD45⁻CD31⁻EpCAM⁺CD104⁺), T cells (PI⁻CD45⁺SSCloFSCloCD3⁺); CD4⁺ T cells (PI⁻CD45⁺SSCloFSCloCD3⁺CD4⁺); CD8⁺ T cells (PI⁻CD45⁺SSCloFSCloCD3⁺CD8⁺); B cells (PI⁻CD45⁺SSCloFSCloCD19⁺); NK cells (PI⁻CD45⁺SSCloFSCloCD3⁻NKp46⁺DX5⁺); Monocytes (PI⁻CD45⁺CD11b⁺SSCloLy6G⁻); Neutrophils (PI⁻CD45⁺CD11b⁺Ly6G⁺); Eosinophils (PI⁻CD45⁺CD11b⁺SSChiLy6G⁻); Alveolar Macrophages (PI⁻CD45⁺CD11c⁺CD103⁻); CD103⁺ Dendritic Cells (PI⁻CD45⁺CD11c⁺CD103⁺). Flow cytometry was performed on the LSR II flow cytometer (BD Biosciences) and data were analyzed using FlowJo software (FlowJo LLC). Fluorescence activated cell sorting (FACS) was performed on the ARIA flow cytometer (BD Biosciences).

**Cell culture.** Single cell suspensions of primary tumor cells were seeded in tumor cell medium (DMEM-F12 + GlutaMAX (Gibco), 10 % FBS (Sigma-Aldrich), 100 U/mL penicillin (Gibco), 100 μg/mL streptomycin (Gibco), 0.04 mg/mL hydrocortisone (Sigma-Aldrich), 1 X Insulin-Transferrin-Selenium-Ethanolamine (Gibco), 5 ng/mL epidermal growth factor (Sigma-Aldrich). Supernatant from the first 6 h of culture was collected for TNF (eBioscience, #5017331) and IL-6 (eBioscience, #501128696) ELISA according to the manufacturer's instructions. The human lung cancer derived cell lines A549, H460, H441 and H358 were obtained from ATCC and cultured in RPMI-1640 + GlutaMAX medium (Gibco) supplemented with 10 % FBS (Sigma-Aldrich), 100 U/mL penicillin 100 μg/mL streptomycin (Gibco). MTS (CellTiter96 Aqueous Non-Radioactive Cell Proliferation Assay, Promega) absorbance assay was used to measure growth of 2000 cells analyzed after 120 h to determine relative growth of human cell lines.

**Reactive oxygen species assay.** 2,7-dichlorofluorescin diacetate (DCFDA) ROS detection assay (Abcam #Ab113851) was performed according to manufacturer's protocol in opaque-walled 96-well plates (Greiner Bio-One). Freshly sorted macrophages, or KK and KP primary tumor cell lines were seeded at 60,000 cells/well, treated with 20 μM DCFDA at 37 °C for 30 min, washed, then treated with 110 μM tert-butyl hydrogen peroxide (TBHP) at 37 °C for 2 h. Fluorescence at excitation/emission 484 nm/535 nm was measured using a Hidex Chameleon microplate reader (LabLogic).

**Seahorse assay.** Single cell suspensions were seeded onto 2 mg/mL type 1 collagen (Corning) coated Seahorse XF96 cell culture microplates (Agilent) at 120,000 cells/well (freshly sorted macrophages or EpCAM⁺ cells). Samples were centrifuged at 400 g at 4 °C for 15 min without brake, then incubated at 37 °C for 1 h. Extracellular acidification rate (ECAR) or oxygen consumption rate (OCR) were measured by XF96 Extracellular Flux Analyzer (Agilent) using serial injections. ECAR: 5 mM glucose, 1 μM oligomycin and 50 mM 2-deoxy-D-glucose (2-DG). OCR:

1 µM oligomycin, 0.7 µM FCCP (carbonyl cyanide-p-tri-fluoromethoxyphenylhydrazone) and 1 µM rotenone and antimycin A.

**Gap19 treatment study**. C57BL/6 mice were subject to intranasal inhalation of 50 µL 2 mg/mL Gap19 peptide (Tocris Bioscience #5353) or PBS vehicle, and collected at 3 and 24 h following infection. Bronchiolar lavage of PBS was passed through the lungs to clear unattached cells, and remaining tissue was prepared for flow cytometry, as above.

**Clodronate liposome treatment study**. KP mice were subject to intranasal inhalation of Ad5-CMV-Cre at 6–8 weeks of age, and randomized into control PBS-liposomes (FormuMax #F70101-NH) or clodronate-liposomes (FormuMax F70101C-NH) four weeks post Ad5-CMV-Cre. Liposomes were administered as 50 µL intranasal injections of 7 mg/mL twice per week for the first week, and weekly thereafter for 7 subsequent weeks. At 12 weeks post Ad5-CMV-Cre, mice were harvested, bronchiolar lavage collected, and lungs processed for histology and flow cytometry, as above.

**6-AN treatment study**. For in vitro single dose studies, A549, H460, H358 and H441 cell lines were treated with 62.5 µM 6-AN (Cayman Chemicals #10009315) for 72 h, and cell survival was then measured using the MTS assay (CellTiter96 Aqueous Non-Radioactive Cell Proliferation Assay, Promega) relative to DMSO vehicle treated cells. Crystal violet colony-forming assay was performed as previously described[57]. For in vivo studies, 6-AN was administered intra-peritoneally at 20 mg/kg body weight once every ten days. Tumor xenografts were generated by subcutaneous transplant of 500,000 cells in 50 % growth factor reduced Matrigel (BD Biosciences) into NSG immune-deficient mice, with treatment commencing once tumors reached a volume of 70–100 mm$^3$.

**Histology and immunohistochemistry**. Lungs were perfused and fixed in 4% paraformaldehyde for 24 h at 4 °C and embedded in paraffin. Sections 2 µm thick were stained with haematoxylin and eosin (H&E) and sections 4 µm thick were immunostained according to standard procedures. Primary antibodies (Supplementary Table 3) were applied overnight at 4 °C in a humidifier box. Slides were imaged using Nikon Eclipse 50i microscope with Axiovision software (Zeiss). Hyperplasia and airway area were quantified using the measure tool on Image J software (Softonic).

**Quantitative RT-PCR**. cDNA was generated from ground lung, tumor tissue or snap-frozen cell pellets using the RNeasy RNA extraction kit (Qiagen), followed by the SuperScript III kit (Thermo Fisher). Quantitative RT-PCR was performed using SyberGreen (Bioline QT605-05) on the Viia7 Real-Time PCR System (Thermo Scientific). Relative mRNA was calculated compared to *Gapdh* internal control using the delta-delta-cT statistical method (primers in Supplementary Table 2).

**Western blotting**. Cell pellets were lysed in RIPA buffer (2% Triton X-100, 0.2% sodium dodecyl sulfate, 0.02% sodium deoxycholate, 0.3 M NaCl, 20 mM Tris hydrochloride, 0.02% sodium azide, complete mini EDTA-free protease inhibitor cocktail (Roche)), and western blotting performed by standard procedures. Primary antibodies (Supplementary Table 3) were incubated overnight at 4 °C and blots exposed using ECL Prime reagent (GE Healthcare) and imaged using Chemidoc Touch (Bio-Rad).

**Analysis for TCGA LUAD *KRAS*-mutated samples**. Gene-wise RNA-seq read counts for lung adenocarcinomas from The Cancer Genome Atlas (TCGA) project[4] were retrieved using the RTCGAToolbox package[58]. One hundred and sixty-two samples from the 20,160,128 release were further stratified by their *KRAS* mutation status[14] leaving 35 samples which were stratified into three groups of similar size based on either their NRF2 signature score (this gene signature was defined by taking the intersection of NRF2 gene lists from Romero et al. (Supplementary Table 3)[18], DeNicola et al. (Supplementary Fig. 8)[53], and Goldstein et al. (Supplementary Table S2)[59], calculating the mean log$_2$ counts per million (log-CPM) for these genes per sample and splitting the samples into low (n = 12), medium (n = 11) and high (n = 12) expression groups) or their NQO1 expression levels (low (n = 12), medium (n = 11) and high (n = 12) expression). Differential expression was assessed for the high vs low groups using limma-voom[60,61] with robust fitting option after trimmed mean of *M*-values (TMM) normalization[62] using edgeR[63]. Adjustment for multiple testing was performed using the FDR method[64]. Enrichment of KEGG metabolism pathways was tested for the high vs low comparison using EGSEA[65] with 10 methods (camera, safe, gage, padog, plage, zscore, gsva, ssgsea, globaltest, and ora) and the median rank to prioritize results. The pheatmap (https://CRAN.R-project.org/package = pheatmap) R package was used to generate heatmaps of expression quantified as log counts per million (log-CPM) values for genes from selected pathways. Software package details can be found in Supplementary Table 4.

**Analysis of mouse RNA-seq data**. RNA-seq read count data were obtain from GEO series GSE83991[38]. Filtering genes with low counts (a count per million greater than one in at least three samples was required) was followed by normalization using the TMM method in the edgeR package[63]. Differential expression analyses used limma-voom[60,61] together with TREAT[66] to test relative to a fold-change cut-off of 1 and multiple testing correction using the FDR method[64]. The NRF2 gene signature defined above was converted from human to mouse using biomaRt[67] and a limma barcode enrichment plot was generated using moderated *t*-statistics[68]. Gene set testing of MSigDB's[69] c5 (Gene Ontology) collection was performed using EGSEA[65] using the 11 methods (camera, safe, gage, padog, plage, zscore, gsva, ssgsea, globaltest, fry, and ora) and the median rank to prioritize results. Differential expression testing of custom gene signatures was performed using ROAST[70].

**Statistics**. Statistical analysis was performed using Prism software (GraphPad Software). Aggressive score was determined as an average of the numerical score attributed to stage. Pairwise comparisons were performed using an unpaired Student *t* test and multivariate comparisons were performed using one-way ANOVA (Kruskal–Wallis test) with Tukey's multiple comparisons test or two-way ANOVA with Dunn's multiple comparisons test for grouped analyses.

**Reporting summary**. Further information on research design is available in the Nature Research Reporting Summary linked to this article.

## Data availability

All data that support the findings of this study is available in the Article, Supplementary Information or available upon reasonable request from the corresponding author. Previously published datasets used in this study are available at Gene Expression Omnibus GSE83991 and through dbGAP with accession #PHS000178.v10.p8 (https://portal.gdc.cancer.gov/projects/TCGA-LUAD).

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

## Acknowledgements

We are grateful to S. Oliver, C. Alvarado, L. Scott, S. O'Connor for animal husbandry, S. Monard in the WEHI Flow Cytometry Facility and E. Tsui in the WEHI Histology Facility for expert support. We acknowledge P. Maltezos in the WEHI Graphics Department for the design and preparation of figure illustrations. We thank R.K. Thomas (Department of Translational. Genomics, University of Cologne) for sequencing data for the Melbourne cohort of the CLCGP. We are thankful to A. Berns, A. Strasser and J. Vissers for critical reading of the manuscript and J. Visvader for useful discussions. This work was supported by an Australian National Health and Medical Research Council (NHMRC) Project Grant to K.D.S. and M.E.R. (1138275), S.A.B (1159002), and Career Development Fellowship to M.E.R. (1104924). K.D.S. is supported by a Victorian Cancer Agency Mid-Career Research Fellowship (18003) and the Peter and Julie Alston Centenary Fellowship; S.D. was supported by an Alan W. Harris Scholarship. This work was made possible through Victorian Government Operational Infrastructure Support and Australian Government.

## Author contributions

The experiments were conceived and designed by S.A.B. and K.D.S. Experiments were performed mainly by S.A.B. with the assistance of A.K., S.D., K.L., B.R., and J.V. Bioinformatic analysis was performed by Y.X. and X.D. under the supervision of M.E.R. Histological examination was performed by V.R. and J.-Y.S. and patient clinical and genomic data was curated by V.R. and G.M.W. The manuscript was written by S.A.B. and K.D.S.

## Additional information

**Competing interests:** The authors declare no competing interests.

