## [Peer Review File · Nature Communications]

REVIEWERS' COMMENTS:

Reviewer #1 (Remarks to the Author):

The authors have satisfactorily addressed my concerns. They have done a nice job addressing the the overlap with Kras-Lkb1 mutant lung cancer and including more convincing data that the findings are indeed specific to Keap1 loss. The manuscript is suitable for publication in Nature Communications in my opinion.

Reviewer #3 (Remarks to the Author):

In the revised manuscript, the authors chose to focus on certain issues raised by other referees and not work to improve the metabolism aspect of the paper. The evidence for the pentose phosphate pathway representing a specific vulnerability of Keap1 mutant lung cancer is still not very compelling. The lack of genetic data to support conclusions drawn with 6-AN is particularly disappointing and will limit the impact of the study. With that said, as stated previously, aspects of the paper are interesting and should be published.

1. The authors provided a referee figure to address the concern that differences in age-matched KK, KP and K tumors might be driven by difference in grade or stage of tumorigenesis. While the data in the referee figure might argue differences in early macrophage infiltration can be observed at the time of tumorigenesis speaks to that phenotype, why not show this data and present that argument in the paper?

2. Similarly, the comment about evidence for KK tumors arising in bronchiolar cells being indirect is addressed with quantification of alveolar and bronchiolar hyperplasia in KP and KK lungs three weeks following Ad5-CMV-Cre infection and also shown in a referee figure. Why not also present this data in the paper?

3. Regardless of what is published, 6-AN is widely recognized to not be specific and anyone serious about pharmacology of inhibiting metabolism will not be impressed by these data. Validating their findings with genetic tools is a straightforward experiment and should be done for publication in a top journal.

4. The metabolism aspects of the paper remain fairly weak and relies on indirect measurements such as ECAR and inferences from transaldolase upregulation. When coupled with the use of a non-specific inhibitor this will limit the overall impact of this part of the study.

Reviewer #4 (Remarks to the Author):

In their manuscript "Unique cellular and metabolic susceptibilities underpin the genetic heterogeneity of KRAS-mutant lung adenocarcinoma", Best et al. evaluated the consequence of concomitant deletion of KEAP1 and expression of K-RasG12D and compared these effects to a other genetic combinations found in lung adenocarcinoma. They find that in contrast to p53 deficient tumors, KEAP1 deficient tumors are bronchiolar in origin and have fewer pro-tumorigenic alveolar macrophages. Further, they are more dependent on the pentose phosphate pathway. This study is much improved and the authors have done a very good job addressing the concerns of the reviewers. Overall, this study is well done and of general interest to the NRF2 community.

However, a few issues remain to be addressed before this study is suitable for publication.

Major comments:

1. The inclusion of the K cohort greatly strengthens the manuscript. However, the inclusion of KPK and KPL mice is very preliminary and does not add much to the overall manuscript.
2. The authors use survival as a readout of the effects of tumor suppressor deletion, but this is used inconsistently throughout the study. Keap1 deletion decreases survival in both the K and KP models but the authors conclude that "Keap1 does not significantly collaborate with p53 or Lkb1 loss to accelerate KrasG12D-induced tumorigenesis" (lines 148-149). Statistical analysis is not included. Tumor burden in the KP model is not assessed, and the number of KPK mice analyzed is quite small (only 5). Therefore, it is unclear whether Keap1 loss cooperates with p53 loss or not.
3. Related to point #1, increased clarity is needed in the discussion regarding the Keap1/Kras/p53 combination (lines 348-353). The authors conclude that the lack of synergy are consistent with the mutual exclusivity of these genetic alterations in LUAD, but (1) it is unclear based on the data presented whether or not synergy is observed and (2) they also observe no synergy in the Keap1/Kras/Lkb1 combination, which are co-mutated, so this conclusion seems incorrect.
4. The discussion of the different models of Keap1 inactivation in LUAD (lines 338-353) would benefit from also mentioning the recent publication of a Keap1 mutant model (Kang. et al. eLife 2019), which found that expression of mutant Keap1 actually decreased tumor formation in the KP model.
5. The inclusion of additional data to support the macrophage data greatly strengthens the manuscript. However, critical information is missing. For instance, in Figure 3c, the authors show that there is a difference in carcinoma incidence between KK and KP mice, and suggest this is related to more macrophages in the KP model. To show causality they treat mice with clodronate, and observe a decrease in tumor size. However, tumor grade and tumor size are different concepts. How does clodronate treatment affect tumor grade?
6. It is unclear what the conclusions are regarding the 6-AN/PPP studies. Do the authors argue that the PPP facilitates redox balance, glycolytic flux, or both? The authors focus on TALDO1 in Figure 6, but 6-AN is an inhibitor of PGD, which is in the oxidative arm and generates NADPH. While glycolysis is assayed, markers of oxidative stress are not evaluated. Do 6-AN treated KPK tumors show more oxidative stress than KP tumors as determined by MDA or 8-oxo-dG IHC (or other markers)? Can the effects be rescued by antioxidant treatment?

Minor points:

1. Line 401 – "glutaminase biosynthesis" – biosynthesis should be removed.
2. Figure 4b - please add the unit of Cx43 expression (relative or absolute).

“Response to referees”
Unique cellular and metabolic susceptibilities underpin the genetic heterogeneity of
***KRAS*-mutant lung adenocarcinoma**
Best *et al.*,

We thank the editor and reviewers for providing valuable feedback on our revised manuscript. The final version of our manuscript has been modified to include the reviewer data as requested by Reviewer #3 (outlined in detail below), and a discussion that highlights the limitations of 6-AN. Furthermore, we have reworded the title to deemphasise the metabolic vulnerability and to avoid the use of the word unique. The title now reads “Distinct initiating events underpin the immune and metabolic heterogeneity of *KRAS*-mutant lung adenocarcinomas”. We believe this title encapsulates all the major findings of our study.

The remaining concerns raised by the reviewers are detailed below:

Reviewer #1 (Expertise: Kras-driven lung cancers):

No comments to address.

Reviewer #3 (Expertise: Cancer Metabolism):

The authors provided a referee figure to address the concern that differences in age-matched KK, KP and K tumors might be driven by difference in grade or stage of tumorigenesis. While the data in the referee figure might argue differences in early macrophage infiltration can be observed at the time of tumorigenesis speaks to that phenotype, why not show this data and present that argument in the paper?

As suggested by the reviewer, the revised manuscript has been updated to include the H&E-stained sections from the lungs of KP and KK mice that depict the differential infiltration of alveolar macrophages seen in early hyperplastic lesions (**Supplementary Fig. 3b**).

Similarly, the comment about evidence for KK tumors arising in bronchiolar cells being indirect is addressed with quantification of alveolar and bronchiolar hyperplasia in KP and KK lungs three weeks following Ad5-CMV-Cre infection and also shown in a referee figure. Why not also present this data in the paper?

We refer to reviewer to **Supplementary Fig. 5c-d** of the revised manuscript, which depicts the quantification of alveolar and bronchiolar hyperplasia in KP and KK lungs three weeks following Ad5-CMV-Cre infection. We have further modified the figure to include representative H&E-stained sections from KP and KK lungs analysed 3 weeks following Ad5-CMV-Cre infection (**Supplementary Fig. 5b**).

Regardless of what is published, 6-AN is widely recognized to not be specific and anyone serious about pharmacology of inhibiting metabolism will not be impressed by these data. Validating their findings with genetic tools is a straightforward experiment and should be done for publication in a top journal.

We agree that pharmacological inhibition inevitable coincides with off-target effects. We have now included a statement in the discussion highlights this point raised by the reviewer (lines 402-405). We would however like to emphasize that our studies using 6-AN are consistent

with genetic studies that inhibit both the oxidative and non-oxidative arms of the PPP¹. Together, these findings give us the confidence that the effects we are observing are a result of inhibition of the PPP.

The metabolism aspects of the paper remain fairly weak and relies on indirect measurements such as ECAR and inferences from transaldolase upregulation. When coupled with the use of a non-specific inhibitor this will limit the overall impact of this part of the study.

We refer the reviewer to our above response and reiterate that in the literature, 6-AN is routinely utilized to inhibit the PPP

Reviewer #4 (Expertise: NRF2):

The inclusion of the K cohort greatly strengthens the manuscript. However, the inclusion of KPK and KPL mice is very preliminary and does not add much to the overall manuscript.

One of the strengths of our manuscript is the generation of a series of genetically engineered mouse models (GEMMs) that reflect all key co-mutation combinations in *KRAS*-mutant lung adenocarcinoma. These models will prove to be powerful tools to interrogate outstanding basic biology and clinical translation questions.

The authors use survival as a readout of the effects of tumor suppressor deletion, but this is used inconsistently throughout the study. Keap1 deletion decreases survival in both the K and KP models but the authors conclude that “Keap1 does not significantly collaborate with p53 or Lkb1 loss to accelerate Kras^{G12D}-induced tumorigenesis” (lines 148-149). Statistical analysis is not included. Tumor burden in the KP model is not assessed, and the number of KPK mice analyzed is quite small (only 5). Therefore, it is unclear whether Keap1 loss cooperates with p53 loss or not.

We now provide statistical analysis of the survival rates of KP versus KPK (Mantel-Cox test $p = 0.36$) and KL versus KLK (Mantel-Cox test $p > 0.99$) on lines 179-182 of the manuscript. Indeed, this data suggests that *Keap1* loss does not significantly collaborate with *p53* or *Lkb1* loss to accelerate *Kras*^{G12D}-induced tumorigenesis. We have preliminary FACS data to indicate no further epithelial cell expansion in KPK mice, supporting the idea that *Keap1* and *p53* alterations are not synergistic in *Kras*-mutant lung cancer.

Related to point #1, increased clarity is needed in the discussion regarding the Keap1/Kras/p53 combination (lines 348-353). The authors conclude that the lack of synergy are consistent with the mutual exclusivity of these genetic alterations in LUAD, but (1) it is unclear based on the data presented whether or not synergy is observed and (2) they also observe no synergy in the Keap1/Kras/Lkb1 combination, which are co-mutated, so this conclusion seems incorrect.

We apologise for the confusion. We have now included K-M curve survival analysis of KP versus KPK mice following Ad5-CMV-Cre infection in **Supplementary Fig. 2a**. Whilst KPK animal numbers are low ($n=5$) this study revealed no statistically significant difference in the survival of KP mice following *Keap1* loss (Mantel-Cox test $p = 0.36$). The results (lines 180-181) section have been updated to reflect these findings.

To address the second comment, it is correct that we also observed no synergy in the *Kras*/*Lkb1*/*Keap1* (KLK) combination (**Table 1**). We however, cannot rule out that the short

latency of both the KL and KKK models restricts our ability to detect further acceleration following combined loss of *Keap1*. To definitively address this, we would need to infect mice with a lower virus titre to induce transformation in a smaller number of lung epithelial cells. These experiments are currently in progress.

The discussion of the different models of Keap1 inactivation in LUAD (lines 338-353) would benefit from also mentioning the recent publication of a Keap1 mutant model (Kang. et al. eLife 2019), which found that expression of mutant Keap1 actually decreased tumor formation in the KP model.

We thank the reviewer for raising this relevant publication, which was not published at the time of our initial manuscript submission. Kang *et al.*, describe a novel *Keap1* mouse model based on the conditional expression of a mutant form of *Keap1*². We now discuss the findings of this publication in the discussion (lines 424-427).

The inclusion of additional data to support the macrophage data greatly strengthens the manuscript. However, critical information is missing. For instance, in Figure 3c, the authors show that there is a difference in carcinoma incidence between KK and KP mice, and suggest this is related to more macrophages in the KP model. To show causality they treat mice with clodronate, and observe a decrease in tumor size. However, tumor grade and tumor size are different concepts. How does clodronate treatment affect tumor grade?

The reviewer raises an interesting point that was not directly addressed in this study, but forms the basis of future work outside the scope of this manuscript.

It is unclear what the conclusions are regarding the 6-AN/PPP studies. Do the authors argue that the PPP facilitates redox balance, glycolytic flux, or both? The authors focus on TALDO1 in Figure 6, but 6-AN is an inhibitor of PGD, which is in the oxidative arm and generates NADPH. While glycolysis is assayed, markers of oxidative stress are not evaluated. Do 6-AN treated KPK tumors show more oxidative stress than KP tumors as determined by MDA or 8-oxo-dG IHC (or other markers)? Can the effects be rescued by antioxidant treatment?

These studies were not performed and are outside the scope of this manuscript.

Line 401 – “glutaminase biosynthesis” – biosynthesis should be removed.

Biosynthesis has been removed from the text (now line 493).

Figure 4b - please add the unit of Cx43 expression (relative or absolute).

We apologise for this oversight. The y axis on Fig. 4b now reads; Cx43 expression relative to *Gapdh*.

REFERENCE LIST:

- 1 Mitsuishi, Y. *et al.* Nrf2 redirects glucose and glutamine into anabolic pathways in metabolic reprogramming. *Cancer cell* **22**, 66-79, doi:10.1016/j.ccr.2012.05.016 (2012).
- 2 Kang, Y. P. *et al.* Cysteine dioxygenase 1 is a metabolic liability for non-small cell lung cancer. *Elife*, **8** (2019).